# Contemporary Trends and Predictors of pT0 in Radical Cystectomy Specimens Among Non-Muscle and Muscle-Invasive Bladder Cancer Patients: A Propensity Score-Matched Analysis from a Single Tertiary Centre in the United Kingdom

**DOI:** 10.3390/cancers17193110

**Published:** 2025-09-24

**Authors:** Francesco Del Giudice, Valerio Santarelli, Katarina Spurna, Syed Ghazi Ali Kirmani, Noor Huda Bhatti, Yasmin Abu-Ghanem, Elsie Mensah, Benjamin Challacombe, Samuel J. Davies, Mohammad Hegazy, Youssef Ibrahim, Mohammed Gad, Amir Khan, Roberta Corvino, Felice Crocetto, Jan Łaszkiewicz, Bernardo Rocco, Benjamin I. Chung, Ramesh Thuraraja, Muhammad Shamin Khan, Rajesh Nair

**Affiliations:** 1Department of Maternal-Infant and Urological Sciences, “Sapienza” University of Rome, Umberto I Hospital, 00185 Rome, Italy; 2Department of Urology, Stanford University School of Medicine, Stanford, CA 94304, USA; 3Guy’s and St. Thomas’ NHS Foundation Trust, Guy’s Hospital, London SE1 7EH, UK; 4Division of Urology, University of Maryland School of Medicine, Baltimore, MD 21201, USA; 5Department of Neurosciences, Reproductive Sciences and Odontostomatology, University of Naples “Federico II”, 80131 Naples, Italy; 6University Centre of Excellence in Urology, Wroclaw Medical University, 50367 Wroclaw, Poland; 7Department of Urology, Fondazione Policlinico Universitario A. Gemelli IRCCS, Università Cattolica del Sacro Cuore, 00136 Rome, Italy

**Keywords:** bladder cancer, radical cystectomy, downstaging, NMIBC

## Abstract

In bladder cancer patients treated with radical cystectomy with or without neoadjuvant chemotherapy, the achievement of no residual tumour is related to better survival outcomes. The aim of the study was to identify the main clinical and pathological features in this population as it could lead to refine treatment choices including bladder-sparing strategies. We examined patients with both Non Muscle-Invasive and Muscle Invasive Bladder Cancer treated between 2009 and 2024. Findings showed how obtaining a cancer-free status after treatment was more likely in male patients (for NMIBC), undergoing neoadjuvant chemotherapy (for MIBC), and without additional risk factors, such as BCG unresponsiveness and concomitant CIS (for both NMIBC and MIBC). These findings support personalization of care through tailoring of therapeutic choices to patient and tumour characteristics.

## 1. Introduction

Radical cystectomy (RC) is the current gold-standard surgical treatment with or without neoadjuvant chemotherapy (NAC) for muscle-invasive bladder cancer (MIBC). Non-muscle-invasive bladder cancer (NMIBC) accounts for 75% of new BC diagnoses and can effectively be treated conservatively in most cases, with Transurethral Resection of Bladder Tumour (TURBT) followed by adjuvant intravesical chemotherapy or Bacillus Calmette–Guérin (BCG) [1,2]. However, when conservatively managed, up to 45% of high-risk or very-high-risk NMIBC patients may progress to muscle-invasive disease within 5 years [3]. Risk classifications have been used as a useful tool to risk stratify NMIBC, serving as a guide for appropriate treatment recommendations, surveillance schedules, and recruitment into clinical trials [4]. The European Association of Urology (EAU) expanded the spectrum to the ‘very-high-risk’ class in addition to the existing low, intermediate, and high-risk stratifications [5]. These updated prognostic risk groups are based on patients’ and tumour characteristics as well WHO 1973 and WHO 2004/2016 grade classifications, predicting the risk of progression to MIBC at 1, 5, and 10-years. The calculated probability of progression at 10 years is as low as 3% for low-risk disease but as high as 59% for very-high-risk patients. Accordingly, current EAU guidelines include recommendations for offering upfront RC to high-risk patients and strongly recommend it for the very-high-risk group [6]. However, this novel risk stratification is not without limitations. Firstly, it does not take into account the positive prognostic effect of adjuvant BCG treatment, which has been demonstrated to reduce the risk of progression [7]. Secondly in BCG-unresponsive NMIBC, other treatment options, mainly Immune Checkpoint Inhibitors (ICIs), have emerged as valid and safe alternatives to upfront RC [8]. Finally, current risk stratification does not consider additional prognostic features, such as Carcinoma In Situ (CIS) in the prostatic urethra, variant histology (VH), a history of recurrence, and recurrence rate by year [9].

While the absence of residual cancer in RC specimens (pT0) is commonly perceived by urologists as a positive prognostic pathological finding, it raises concerns of overtreatment for NMIBC as RC has serious detrimental impacts on patients’ Quality of Life (QoL) [10,11].

In the case of MIBC, pT0 in RC specimens with or without prior NAC confers an overall and cancer-specific survival benefit [10,12]. However, those with final pT0 staging may be potential candidates for bladder-sparing surgery (BSS). The aim of our study is to analyse preoperative clinical and pathological characteristics from our institutional pT0 cohort of RC patients. The key objective is to identify those variables which increase the likelihood of achieving a pT0 status at final pathology in two distinct cohorts—NMIBC and MIBC. Lastly, we provide a contemporary overview of pT0 trends following RC, with the goal of enhancing patient counselling and refining the prognostic implications associated with this pathological outcome.

## 2. Materials and Methods

### 2.1. Study Cohort

All patients undergoing radical cystectomy at our institution between 2009 and 2024 for either high-/very-high-risk NMIBC or T2N0M0 MIBC with curative intent were retrospectively reviewed from our prospectively maintained institutional RC database. Clinical, anthropometric, and demographic characteristics (i.e., gender, Age-adjusted Charlson Comorbidity Index [ACCI], body mass index [BMI], age, renal function) were recorded. Patients’ management at the time of RC was according to the Trust, National, and EAU Guidelines. The inclusion criteria were as follows: the absence of distant metastases at preoperative clinical staging; histologically confirmed diagnosis of urothelial or mixed histology, primary or recurrent BC; at least one prior TURBT before RC; RC performed as primary treatment option with curative intent; the ability to sign an informed consent for the procedure and the adoption of de-identified data for research purposes. Exclusion criteria were the presence of radiologically suspected or histologically confirmed distant metastases and previous External Beam Pelvic Radiation Therapy (EBRT) or cystectomy (not RC) performed for indications other than BC (e.g., chronic pelvic pain syndrome, radiation cystitis, palliative or salvage intent).

Preoperative local and distant staging was performed with TURBT and a combination of Computerized Tomography (CT), pelvic multiparametric Magnetic Resonance Imaging (mpMRI), or fluorodeoxyglucose positron emission tomography/CT (FDG-PET/CT), as appropriate for indications and contraindications for each diagnostic modality in accordance with Urothelial Multidisciplinary Team (MDT) recommendations. According to MDT recommendations and following the latest EAU guidelines, NAC was offered only to cisplatin-eligible MIBC patients and performed with a combination of cisplatin and at least one additional chemotherapeutic agent. The high risk of disease progression was discussed with high-/very-high-risk NMIBC patients, as well as the possible complications and impact on Quality of Life (QoL) associated with RC, in line with the principles of shared and informed decision-making. In accordance with EAU guidelines, predictors of higher risk of disease progression not included in current risk stratification (such as BCG unresponsiveness and histological variants) were also discussed. In accordance with EAU guidelines, BCG failure broadly refers to any HG disease occurring during or after BCG therapy [13].

### 2.2. Surgical Procedure

All RCs were performed by a team of three experienced consultants (MSK, RT, RN) at the same institution and in the setting of a UK-certified Robotic Clinical Fellowship programme. All patients underwent RC with an open or robotic-assisted approach, carried out with the standardized technique previously described [14]. The diversion of choice (intracorporeal or extracorporeal Ileal Conduit [IC], Orthotopic Neobladder, or Ureterocutaneostomy [UCS]) was selected after adequate patient counselling and MDT assessment. The decision to perform a Pelvic Lymphadenectomy (PLND), with either a standard or extended template, was left to the surgeon’s discretion, in accordance with preoperative staging and MDT discussions.

All RC specimens were analysed by experienced uropathologists with >20 years of experience in urological malignancies. The histological subtype, pathological T (pT) and N (pN) stage, number of lymph nodes, grade, surgical margins, concomitant CIS, and lymph–vascular invasion were recorded in the final histopathological report, in accordance with the American Joint Committee on Cancer (AJCC) guidelines. pT0 status was defined as the absence of residual cancer in the RC specimen.

### 2.3. Statistical Analysis

Statistical analyses were performed with STATA version 18.1 (Stata Corporation, College Station, TX, USA) and SPSS version 27.0 (IBM SPSS Statistics for Windows, Version 27.0. Armonk, NY, USA: IBM Corp.), following the standardized methodology previously described [15]. Descriptive statistics summarized baseline characteristics of the study population. Pertinent continuous variables were reported as the median and interquartile range (IQR) and categorical variables as numbers and percentages. To balance baseline characteristics, a 1:1 propensity score match (PSM) analysis using the nearest-neighbour method with a calliper size of 0.1 was performed between pT0 and non-pT0 patients based on Age-adjusted CCI and preoperative clinical stage (NMIBC vs. MIBC). Subsequently, pertinent study information of the matched cohorts was reported, stratified according to clinical stage (cTis-T1 vs. cT2) and pathological T stage (pT0 or >pT0). Quantitative data and pairwise intergroup comparison of variables was conducted with the Mann–Whitney test or the ANOVA one-way test. Lastly, clinically relevant variables were selected for multivariable logistic regression modelling to identify independent predictors of the absence of residual cancer in RC specimens (pT0) separately for MIBC and NMIBC patient cohorts.

## 3. Results

### 3.1. Description of the Study Cohort

A total of 655 patients, 498 (76%) males and 157 (24%) females, who underwent RC with curative intent at our institution were included the analysis. The general characteristics of the studied population are described in Table 1. The median age, BMI, and ACCI were 69.6 (IQR 62.7–75.3), 27.5 (IQR 62.7–75.3), and 5 (3–6), respectively. Of these, 373 (57%) had high- or very-high-risk NMIBC, while 282 (43%) had MIBC. Of the entire cohort, 117 (17.9%) had pT0 in the RC specimen. The rate of patients achieving a pT0 status was the highest in 2009 (41% of NMIBC and 25% for MIBC) and the lowest in 2018 (4% of NMIBC and 0% of MIBC), showing variable prevalence patterns during the years (Figure 1). Subsequently, 114 of the 117 pT0 patients were matched with patients with 114 >pT0 patients based on the ACCI and according to the original preoperative clinical stage (NMIBC vs. MIBC). A comparison of preoperative, perioperative, and pathological characteristics between pT0 and >pT0 patients is reported in Table 2, stratified by preoperative cT stage group. In the NMIBC group, pT0 patients were more frequently males (*n* = 54 [83.1%] males and *n* = 11 [16.9%] females) when compared to the >pT0 cohort (*n* = 43 [66.2%] males and *n* = 22 [33.8%] females (*p* = 0.027)) and had lower rates of BCG treatment failure (53.8% vs. 46.2%, *p* = 0.018).

In the MIBC group, pT0 patients were significantly older (median age: 70 [IQR 63.5–75.2] vs. 66 [IQR 56.7–74.3], *p* = 0.05) and a higher number had received NAC (*n* = 33 [67.3%] vs. *n* = 22 [44.9%], *p* = 0.04). In both groups, pT0 patients were less likely to have CIS at preoperative TURBT (36.9% vs. 53.8% for NMIBC, *p* = 0.05, and 8.2% vs. 24.5% for MIBC, *p* = 0.029). Moreover, pT0 patients of both groups had lower rates of VH at TURBT specimen, but the difference did not reach significance (*p* = 0.08 and *p* = 0.06), mainly due to the relatively low prevalence of histological variants (only 11.4% of the whole TUR samples). Only one pT0 patient per group had cN+ at final histology.

### 3.2. Predictors of pT0 Status by Clinical Stage Group

Conditional multivariate logistic regression exploring the likelihood of pT0 across the PSM population is shown in Table 3 and Table 4 for NMIBC and MIBC, respectively. In the first group, across the explored variables, male gender was only associated with an increased likelihood of pT0 (aOR: 2.89, 95% CI 1.13–7.30). Conversely, previous BCG treatment failure and additional CIS significantly and independently decreased the likelihood of pT0 in NMIBC patients (aOR: 0.40, 95% CI 0.19–0.92, and OR: 0.16, 95% CI 0.03–0.97). For MIBC patients, only concomitant CIS preoperatively was found to be a negative predictor of final pT0 status (aOR: 0.22, 95% CI 0.06–0.83). On the other hand, NAC more than doubled the likelihood of absence of residual tumour in the final pathological analysis (aOR: 2.20, 95% CI 1.01–6.82, *p* = 0.049). In both groups, the presence of VH at TURBT predicted a detrimental impact on pT0 status, but the impact did not reach statistical significance, very likely due to the fact that only few cases presented with reported preoperative VH across the specific population (aOR: 0.49, 95% CI 0.20–1.08 and aOR: 0.36, 95% CI 0.10–1.21 for NMIBC and MIBC, respectively).

## 4. Discussion

In recent years, BSS has been discussed as a possible alternative to RC in selected cases, provisionally demonstrating similar survival outcomes, but the evidence base is relatively less robust [16]. Hence, RC remains the gold standard to achieve a cure in localized disease. It is also considered a safer option for those patients with poor adherence to follow-up. However, RC is a urological intervention with high morbidity, possibly resulting in a significant decline in patients’ perceived QoL [11,17,18]. Despite recent increased attention towards multimodal rehabilitation programmes, RC negatively impacts urinary, bowel, and sexual function and affects perceived body image, which can lead to psychological sequelae [19,20].

Current guidelines recommend platinum-based NAC for platinum-eligible cT2-T4N0M0 patients [21]; recent evidence confirms the positive prognostic significance of pT0 status after RC and suggests that the survival benefit of NAC is strictly dependent on the increased probability of downstaging to pT0 disease [10,12,22]. Delayed time to RC and suboptimal chemotherapy are the main factors associated with a poor response to NAC, which ultimately affects survival outcomes [23]. A higher tumour grade and cT have also been associated with failure to achieve complete pathological response and an increased risk of later upstaging [14,24]. Regarding VH, studies focusing on the ideal treatment approach are scarce. Some studies show that tumours with variant histologic patterns have a greater pathological response to NAC, while others report lower rates of ypT0 when compared to Pure Urothelial Carcinoma (PUC) [25,26]. The main limitation of currently available studies lies in the tendency to treat histological variants as a single entity. This approach, justified by the low overall prevalence of histological variants, overlooks the distinct microscopic and nuclear features that characterize each variant, which may significantly influence their individual responsiveness to chemotherapy.

The risk of progression for NMIBC is defined by clinical and pathological patient and tumour characteristics. Upfront RC is recommended for high-risk NMIBC and strongly recommended for the very-high-risk group; NAC is not recommended for these sub-groups of patients given the localized nature of the disease [27]. Accordingly, the absence of residual tumour on the bladder specimen is an indicator of an adequate and complete resection. For this group, RC is proposed only in cases of high-risk progression as it has a negative impact on cancer-specific survival.

In this study, we described preoperative and perioperative characteristics of pT0 patients at RC. Additionally, we compared our pT0 cohort with matched >pT0 counterparts to accurately define predictors of pT0 status at RC. In the high-risk and very-high-risk NMIBC group, we found that female sex, the presence of additional CIS, and previous BCG treatment were associated with lower rates of pT0 in RC specimens. Pure or mixed VH showed a clear tendency towards a lower probability of pT0, despite it not reaching significance, very likely due to the relatively small prevalence of this at TUR pathology. Of the aforementioned variables, only concomitant CIS is considered in the currently adopted risk stratification.

To more efficiently identify patients who would clearly benefit from an upfront RC, more comprehensive risk stratification is required. The higher rates of progression to extravesical disease and worse survival outcomes of female patients have already been demonstrated [28,29,30]. Our results, paired with the lower psychological impact of the long-term functional complications of RC, could suggest a lower threshold for proposing definite treatment in female patients compared to male patients [31].

Regarding MIBC, in addition to the well accepted role of NAC in increasing the likelihood of a final pT0 status, we found that the absence of CIS or histological variants could predict a higher probability of no evidence of residual cancer in RC specimens. Moreover, in our cohort, only cT2 patients successfully achieved a pT0 status, specifically in 27 (55%) and in 16 (32.7%) >pT0 and pT0 patients, respectively, among those with MIBC who did not receive NAC. Considering that NAC has been the standard of care for cT2-T4N0M0 since 2008, these rates might seem relatively high. However, the vast majority of our MIBC cohort had organ-confined disease (i.e., cT2N0M0). In this particular group, the role of NAC in reducing cancer-specific mortality (CSM) was recently formally validated, compared to its well established role in non-organ-confined disease [32,33]. Accordingly, following MDT assessment and patient counselling, NAC was omitted in a relevant portion of the study population, in the light of the still uncertain survival benefit and possibly high rate of unfit patients. The high prevalence of cT2N0M0 patients also partially explains the relatively high representation of pT0 patients who did not undergo NAC, since, when possible, an aggressive TUR procedure was performed, particularly in patients with resectable tumours and without a clear preoperative stage. The rate of complete pathological response (pCR) to NAC in our matched cohort (*n* = 33 patients, 60%) was higher than that reported in the literature (30–40%) [34,35]. This discrepancy may be explained by PSM, with 50% of our cohort represented by pT0 patients, and by precise patient selection following MDT discussion and the latest guidelines in our high-volume reference centre. Nonetheless, our aim was not to define predictors of NAC response, but to provide a wider description of the pT0 population. Given the high accuracy of novel diagnostic and follow-up modalities, such as multiparametric Magnetic Resonance Imaging (mpMRI) with VI-RADS score determination and Photodynamic diagnosis (PDD), and given the promising role of Immune Checkpoint Inhibitors (ICIs) for non-metastatic BC and their possible association with EBRT, a deep understanding of pT0 predictors could help in tailoring and integrating future multimodality treatment strategies. These results gain additional interest as BSS is becoming increasingly popular. Understanding predictors of a favourable pathological outcome, particularly for NMIBC patients who did not receive NAC, might raise the threshold for RC consideration in this particular class of patients. This is particularly true as the therapeutic arsenal for NMIBC is expanding to systemic treatment, particularly in BCG-refractory disease [9]. Regarding MIBC, while it is true that regimens adopted during NAC and Trimodality Therapy (TMT) differ, a more thorough understanding of the clinical and pathological characteristics associated with downstaging and pCR after NAC could help better define predictors of favourable outcomes of BSS. As research is focusing on expanding ICI indications from TMT to the NAC setting, factors currently predicting a better response to ICIs might become selection criteria for NAC [36]. Finally, in the context of personalised medicine, a precise classification of predictors of good response to one regimen or another (ICIs vs. platinum-based) might help in offering tailored treatment in the context of NAC and TMT.

The steady temporal trend in pT0 status since 2009 reported in our study, despite constant fluctuations over the years, is confirmation of the lack of clear advancements in BC management in the last 20 years. An increasing rate of pT0 status could have indicated an improvement in earlier tumour detection, TUR precision, and NAC efficacy. On the contrary, a trend towards a reduction in pT0 rates could have suggested an increasing number of BSSs being adopted as a result of precise patient selection criteria for MIBC, and the availability of novel adjuvant therapeutic options for BCG-refractory NMIBC. However, recent and potential future positive results of large multicentre studies, particularly regarding the adoption of ICIs for NMIBC and their introduction into NAC regimens, could help shape future trends of pT0 status [37].

Several limitations of our study must be acknowledged. First, the retrospective nature and the relatively small sample size could have influenced the results. Relevant information, such as tumour size, BCG treatment duration, and NAC regimens, were not available. Although PSM helped in obtaining similar and comparable study groups, this study is still susceptible to potential biases arising from unobserved confounders. Moreover, it could have introduced selection and collider biases with unpredictable effects on the explored variables. Lastly, we did not explore or report survival outcomes and trajectories since this was not in the original scope of the present analysis.

## 5. Conclusions

In the present study, we explored temporal trends and predictors of pT0 status in a cohort of high- and very-high-risk NMIBC and MIBC patients. In addition to the already accepted negative predictive role of additional CIS, we found female sex and BCG failure to be independently associated with a lower likelihood of pT0 status in the NMIBC cohort. These findings may improve the accuracy of current risk stratification models by refining NMIBC patient selection and enhancing the prediction of pathological outcomes. Regarding MIBC, cT2N0M0 without additional risk factors (concomitant CIS or VH) could benefit from re-evaluation with novel diagnostic techniques (mpMRI and Re-TURBT with PDD), before or after chemotherapy, to select those patients with a higher success probability using a bladder-sparing approach. Finally, while BC with histological variants has already been associated with a poorer prognosis, in the future, specific histological subtypes should be evaluated as a separate entity to allow for personalisation of individual care.

## Figures and Tables

**Figure 1 cancers-17-03110-f001:**
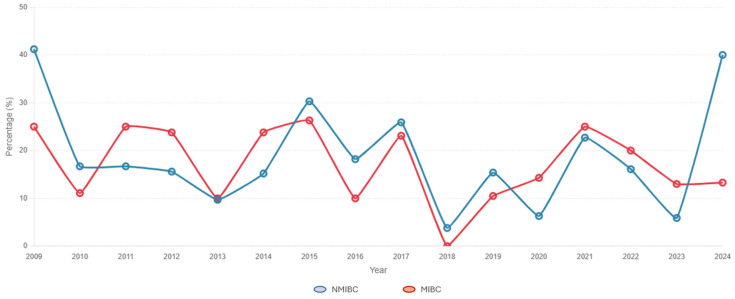
Rates of pT0 achievement by year across *n* = 655 radical cystectomies with curative intent performed between 2009 and 2024. **NMIBC**: non-muscle-invasive bladder cancer; **MIBC**: muscle-invasive bladder cancer.

**Table 1 cancers-17-03110-t001:** The baseline characteristics of the study population before propensity score matching. **BMI**: body mass index; **ACCI**: Age-adjusted Charlson Comorbidity Index; **EGFR**: Estimated Glomerular Filtration Rate; **MDM**: Multidisciplinary Meeting; **CT**: Computerized Tomography; **MRI**: Magnetic Resonance Imaging; **LG**: low grade; **HG**: high grade; **BCG**: Bacillus Calmette–Guérin; **NAC**: neoadjuvant chemotherapy.

Variable	Total *N* = 655
**Sex**	
-Male	498 (76%)
-Female	157 (24%)
**Age**	69.6 (62.7–75.3)
**BMI**	27.5 (24.2–30.7)
**ACCI**	5 (3–6)
**Ethnicity**	
-White	508 (77.53%)
-Black	28 (4.3%)
-Asian	21 (3.21%)
-Other	98 (14.96%)
**EGFR**	73 (60–86)
**Creatinine (mg/L)**	86 (75–102)
**Discussed in Pelvic MDM**	
-No	255 (39%)
-Yes	400 (61%)
**CT**	
-No	28 (4.3%)
-Yes	627 (95.7%)
**MRI**	
-No	563 (86%)
-Yes	92 (14%)
**Grade**	
-LG	34 (5.2%)
-HG	621 (94.8%)
**MIBC vs. NMIBC**	
-NMIBC	373 (57%)
-MIBC	282 (43%)
**Clinical T Stage**	
-T0	3 (0.5%)
-Ta	74 (11.3%)
-Tis	30 (4.6%)
-T1	266 (40.5%)
-T2	272 (41.5%)
-T3	9 (1.4%)
-T4	1 (0.2%)
**Clinical N Stage**	
-Nx	62 (9.5%)
-N0	586 (89.5%)
-N1	7 (1%)
**BCG Failure**	
-No	355 (54.2%)
-Yes	300 (45.8%)
**NAC**	
-No	493 (75.3%)
-Yes	162 (24.7%)
**Pathological T Stage**	
-T0	117 (17.9%)
->T0	538 (82.1%)

**Table 2 cancers-17-03110-t002:** General characteristics based on preoperative stage category (NMIBC vs. MIBC) and final pathological T Stage (T0 vs. >pT0). **BMI**: body mass index; **ACCI**: Age-adjusted Charlson Comorbidity Index; **EGFR**: Estimated Glomerular Filtration Rate; **MDM**: Multidisciplinary Meeting; **BCG**: Bacillus Calmette–Guérin; **UCS**: Ureterocutaneostomy; **IC**: Ileal Conduit; **CT**: Computerized Tomography; **MRI**: Magnetic Resonance Imaging; **PET CT**: positron emission tomography CT; **LG**: low grade; **HG**: high grade; **CIS**: Carcinoma In Situ; **NAC**: neoadjuvant chemotherapy; **OT**: operative time; **EBL**: estimated blood loss; **LOS**: length of stay.

Variable	Total *n* = 228	NMIBC *n* = 130			MIBC *n* = 98		
		>pT0 *n* = 65	pT0 *n* = 65	*p* Value	>pT0 *n* = 49	pT0 *n* = 49	*p* Value
**Sex**				**0.027**			0.170
-Male	169 (74.1%)	43 (66.2%)	54 (83.1%)		33 (67.3%)	39 (79.6%)	
-Female	59 (25.9%)	22 (33.8%)	11 (16.9%)		16 (32.7%)	10 (20.4%)	
**Ethnicity**				0.4			0.5
-White	182 (79.8%)	51 (78.5%)	52 (80%)		37 (73.5%)	42 (85.7%)	
-Black	10 (4.4%)	2 (3.1%)	4 (6.2%)		3 (6.1%)	1 (2%)	
-Asian	7 (3.1%)	1 (1.5%)	4 (6.2%)		1 (2%)	1 (2%)	
-Other	29 (12.7%)	11 (16.9%)	5 (7.7%)		8 (16.3%)	5 (10.2%)	
**Age**	69.3 (62.7–75.1)	69.1 (62.3–76.8)	70.6 (63.2–76.1)	0.7	66 (56.7–74.3)	70 (63.5–75.2)	**0.05**
**BMI**	27.5 (24–30.7)	27.7 (23.7–32.3)	27.4 (24.7–29.9)	0.44	27.2 (23.4–29.9)	27.3 (24–32.5)	0.17
**ACCI**	5 (3–6)	5 (2.5–6)	5 (4–6)	0.75	5 (2–6)	5 (3–6.26)	0.6
**EGFR**	75 (62.5–87)	73 (63–86.5)	77 (63–98.5)	0.13	75 (66–86.75)	71 (55–87)	0.07
**Creatinine (mg/L)**	84 (73–96)	87 (77–96)	83 (67–91)	**0.04**	83.5 (75–96.25)	90 (69–106)	0.18
**Discussed in Pelvic MDM**				0.16			0.5
-No	64 (28.1%)	14 (21.5%)	21 (32.3%)		16 (32.7%)	13 (26.5%)	
-Yes	164 (71.9%)	51 (78.5%)	44 (67.7%)		33 (67.3%)	36 (73.5%)	
**BCG failure**				**0.018**			0.18
-No	117 (51.3%)	17 (26.2%)	30 (46.2%)		32 (65.3%)	38 (71.4%)	
-Yes	111 (48.7%)	48 (73.8%)	35 (53.8%)		17 (34.7%)	11 (22.4%)	
**NAC**				0.3			**0.04**
-No	169 (74.1%)	64 (98.5%)	62 (95.4%)		27 (55.1%)	16 (32.7%)	
-Yes	59 (25.9%)	1 (1.5%)	3 (4.6%)		22 (44.9%)	33 (67.3%)	
**Technique**				0.16			0.6
-Open	61 (26.8%)	21 (32.3%)	14 (21.5%)		13 (26.5%)	13 (26.5%)	
-Robotic	166 (72.8%)	44 (67.7%)	51 (78.5%)		36 (73.5%)	35 (71.4%)	
Laparoscopic	1 (0.4%)	0	0		0	1 (2%)	
**Diversion Type**				0.56			**0.045**
-UCS	1 (0.4%)	1 (1.5%)	0				
-IC	210 (92.1%)	61 (93.8%)	4 (6.2%)		47 (95.9%)	41 (83.7%)	
-Neobladder	17 (7.5%)	3 (4.6%)	61 (93.8%)		2 (4.1%)	8 (16.3%)	
**Lymph Node Excision Technique**				0.5			0.2
-Not Performed	17 (7.5%)	5 (7.7%)	9 (13.8%)		3 (6.1%)	0	
-Standard	166 (72.8%)	48 (73.8%)	44 (67.7%)		36 (73.5%)	38 (77.6%)	
-Extended	45 (19.7%)	12 (18.5%)	12 (18.5%)		10 (20.4%)	11 (22.4%)	
**Clinical T Stage**							**0.004**
-Ta	35 (15.4%)	16 (24.6%)	19 (29.2%)		0	0	
-Tis	8 (3.5%)	7 (10.8%)	1 (1.5%)		0	0	
-T1	87 (38.1%)	42 (64.6%)	45 (67.7%)		0	0	
-T2	88 (38.6%)	0			39 (79.6%)	49 (100%)	
-T3	9 (3.9%)	0			9 (18.4%)	0	
-T4	1 (0.4%)	0			1 (2%)	0	
**Clinical N Stage**				0.29			0.6
-Nx	24 (10.4%)	10 (15.4%)	6 (9.2%)		4 (8.2%)	4 (8.2%)	
-N0	202 (88.6%)	55 (84.6%)	58 (89.2%)		45 (91.8%)	44 (89.8%)	
-N1	3 (0.9%)	0	1 (1.5%)		0	1 (2%)	
**Grade**				0.19			0.15
-LG	12 (5.3%)	3 (4.6%)	7 (10.8%)		0	2 (4.1%)	
-HG	216 (94.7%)	62 (95.4%)	58 (89.2%)		49 (100%)	47 (98%)	
**Histological Variant**				0.08			0.06
-No	202 (88.6%)	58 (89.2%)	63 (96.9%)		37 (75.5%)	44 (89.8%)	
-Yes	26 (11.4%)	7 (10.8%)	2 (3.1%)		12 (24.5%)	5 (10.2%)	
**Additional CIS**				**0.05**			**0.029**
-No	153 (67.1%)	30 (46.2%)	41 (63.1%)		37 (75.5%)	45 (91.8%)	
-Yes	75 (32.9%)	35 (53.8%)	24 (36.9%)		12 (24.5%)	4 (8.2%)	
**pT Stage**				**<0.001**			**<0.001**
-T0	114 (50%)	0	65 (100%)		0	49 (100%)	
-Ta	15 (6.6%)	14 (21.5%)	0		1 (2%)	0	
-Tis	20 (8.8%)	17 (26.2%)	0		3 (6.1%)	0	
-T1	21 (9.2%)	18 (27.7%)	0		3 (6.1%)	0	
-T2	22 (9.6%)	9 (13.8%)	0		13 (26.5%)	0	
-T3	28 (12.3%)	4 (6.2%)	0		24 (49%)	0	
-T4	8 (3.5%)	3 (4.6%)	0		5 (10.2%)	0	
**pN Stage**				0.5			**0.03**
-Nx	17 (7.5%)	5 (7.7%)	6 (9.2%)		4 (8.2%)	2 (4.1%)	
-N0	191 (83.7%)	57 (87.7%)	58 (89.2%)		30 (61.2%)	46 (93.8%)	
-N1	10 (4.0%)	2 (3.1%)	1 (1.5%)		6 (12.2%)	1 (2%)	
-N2	9 (3.9%)	1 (1.5%)	0		8 (16.3%)	0	
-N3	1 (0.4%)	0	0		1 (2%)	0	
**Lymph Node Yield**	17 (12–22)	17 (13–23)	15 (10–22)	0.3	17 (12–21)	17.5 (12.7–21.2)	0.7
**Positive Surgical Margins**				0.3			**0.006**
-No	220 (96.5%)	64 (98.5%)	65 (100%)		42 (85.7%)	49 (100%)	
-Yes	8 (3.5%)	1 (1.5%)	0		7 (14.3%)	0	

**Table 3 cancers-17-03110-t003:** Multivariate regression exploring predictors of T0 in NMIBC patients. **aOR**: adjusted odds ratio; **CI**: Confidence Interval; **MDM**: Multidisciplinary Meeting; **BCG**: Bacillus Calmette–Guérin; **CIS**: Carcinoma in Situ.

Variable	aOR	95% CI	*p* Value
**Male gender**	2.89	1.13–7.30	**0.026**
**Discussed in Pelvic MDM**	0.71	0.32–1.4	0.3
**BCG Failure**	0.40	0.19–0.99	**0.05**
**Additional CIS**	0.16	0.025–0.97	**0.04**
**Histological variant**	0.49	0.21–1.08	0.08

**Table 4 cancers-17-03110-t004:** Multivariate regression exploring predictors of T0 in MIBC patients. **aOR**: adjusted odds ratio; **CI**: Confidence Interval; **MDM**: Multidisciplinary Meeting; **CIS**: Carcinoma in Situ.

Variable	aOR	95% CI	*p* Value
**Male gender**	1.97	0.74–5.25	0.26
**Discussed in Pelvic MDM**	1.14	0.41–3.16	0.8
**Neoadjuvant Chemotherapy**	2.20	1.01–6.82	**0.049**
**Additional CIS**	0.22	0.06–0.80	**0.02**
**Histological variant**	0.36	0.1–1.21	0.09

## Data Availability

The data presented in this study are available in this article.

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
