# Peer review of "Contemporary Trends and Predictors of pT0 in Radical Cystectomy Specimens Among Non-Muscle and Muscle-Invasive Bladder Cancer Patients: A Propensity Score-Matched Analysis from a Single Tertiary Centre in the United Kingdom"

_cancers, 2025, doi:10.3390/cancers17193110_

Round 1

Reviewer 1 Report

Comments and Suggestions for Authors

The authors investigated predictive factors for pT0 status following radical cystectomy (RC) for NMIBC and NIBC. Achieving pT0 indicates a favorable prognosis, and this study is considered meaningful for predicting the efficacy of RC. However, this study has some problems and needs to be improved.

Specific comments:
1.    This study performed a relatively high number of radical cystectomies for NMIBC. While EAU guidelines recommend RC for BCG-refractory NMIBC, this study appears to include many NMIBC patients who underwent RC but were not BCG-refractory. Since urinary diversion significantly impacts QOL, accurate preoperative diagnosis and careful consideration are essential when determining RC eligibility. The criteria for RC should be clarified, particularly for NMIBC.
2.    Why does the MIBC group include a significant number of cases classified as pT0 despite not undergoing NAC? Does this indicate aggressive TUR procedures? Isn't this overdiagnosis? The basis for diagnosing MIBC should be clarified.
3.    NAC plays an important role in achieving pT0 in MIBC. Further details about NAC are needed.
4.    If possible, please indicate whether recurrence occurred in the pT0 cases in this study. Furthermore, even if pT0 is not achieved, a favorable prognosis is considered possible when RC is performed for NMIBC. Was there a difference in recurrence rates between pT0 and >pT0 cases in patients who underwent RC for NMIBC?
5.    Table 1 shows that RC was also performed in cT0 and cTa cases. Specifically, why was RC indicated for cT0? Please explain.
6.    The creatinine range in Table 1 is incorrect. Please verify and include the units.
7.    There are misaligned numbers in the Clinical Stage column of Table 2. Additionally, some parentheses are missing in the Surgical Margin column. Please verify.

Author Response

We sincerely thank you for taking the time to read and evaluate our manuscript and for your useful contribution. We are happy to address your comments.

Reviewer 2 Report

Comments and Suggestions for Authors

see attached

Author Response

We sincerely appreciate the detailed review of our manuscript you provided and thank you for your contribution. When possible, we took care of modifying our paper according to your suggestions, in particular:

Reviewer 3 Report

Comments and Suggestions for Authors

The reviewer appreciates the opportunity to evaluate this intriguing study. Investigating the predictors of pT0 after radical cystectomy is important for refining the criteria for bladder-sparing treatment. It was interesting that the authors separated the two groups, NMIBC and MIBC, as each subtype requires a different treatment strategy.

However, there are several concerns that may affect the interpretation of the results. The completeness of resection at TUR-Bt could be a major factor in predicting pT0 in the final specimen. Did all patients undergo complete resection at TUR-Bt, or were some cases limited to tissue sampling only? This distinction could significantly influence the findings.

In addition, tumor size—which could impact the adequacy of TUR-Bt—should be considered in the analysis, or at least clarified in the table summarizing patient characteristics

Author Response

We appreciate you taking the time to read and evaluate our work, and for your useful comments. We hope you will find our responses fulfilling

Round 2

Reviewer 1 Report

Comments and Suggestions for Authors

The manuscript has been substantially revised and improved compared to the previous version.

Reviewer 2 Report

Comments and Suggestions for Authors

I have no comments

Reviewer 3 Report

Comments and Suggestions for Authors

The authors already addressed all the points the reviewer had suggested.